# Inflammation produces catecholamine resistance in obesity via activation of PDE3B by the protein kinases IKKε and TBK1

Jonathan Mowers[1,2,3†], Maeran Uhm[1,2,3†], Shannon M Reilly[1], Joshua Simon[1], Dara Leto[1‡a], Shian-Huey Chiang[1‡b], Louise Chang[1], Alan R Saltiel[1,2,3]*

[1]Life Sciences Institute, University of Michigan, Ann Arbor, United States;
[2]Department of Internal Medicine, University of Michigan, Ann Arbor, United States;
[3]Department of Molecular and Integrative Physiology, University of Michigan, Ann Arbor, United States

*For correspondence: saltiel@umich.edu

†These authors contributed equally to this work

‡Present address: [a]Department of Biology, Stanford University, Stanford, United States; [b]Muscle Metabolism Discovery and Performance Unit, GlaxoSmithKline, Research Triangle Park, Durham, United States

**Competing interests:** The authors declare that no competing interests exist.

**Abstract** Obesity produces a chronic inflammatory state involving the NFκB pathway, resulting in persistent elevation of the noncanonical IκB kinases IKKε and TBK1. In this study, we report that these kinases attenuate β-adrenergic signaling in white adipose tissue. Treatment of 3T3-L1 adipocytes with specific inhibitors of these kinases restored β-adrenergic signaling and lipolysis attenuated by TNFα and Poly (I:C). Conversely, overexpression of the kinases reduced induction of *Ucp1*, lipolysis, cAMP levels, and phosphorylation of hormone sensitive lipase in response to isoproterenol or forskolin. Noncanonical IKKs reduce catecholamine sensitivity by phosphorylating and activating the major adipocyte phosphodiesterase PDE3B. *In vivo* inhibition of these kinases by treatment of obese mice with the drug amlexanox reversed obesity-induced catecholamine resistance, and restored PKA signaling in response to injection of a β-3 adrenergic agonist. These studies suggest that by reducing production of cAMP in adipocytes, IKKε and TBK1 may contribute to the repression of energy expenditure during obesity.

## Introduction

Obesity generates a state of chronic, low-grade inflammation in liver and adipose tissue accompanied by macrophage infiltration and the local secretion of inflammatory cytokines and chemokines that attenuate insulin action, resulting in insulin resistance and the subsequent development of Type 2 diabetes (*Wellen and Hotamisligil, 2005*; *Hotamisligil, 2006*; *Lumeng et al., 2007*; *Shoelson et al., 2007*). Numerous studies indicate a strong correlation between inflammation and insulin resistance across several populations (*Hotamisligil, 2006*). Moreover, genetic ablation or pharmacological inhibition of inflammatory pathways can dissociate obesity from insulin resistance (*Hotamisligil, 2006*; *Shoelson et al., 2007*), suggesting that local inflammation can be a key step in the generation of insulin resistance.

The transcription factor NFκB and its inflammatory program play an important role in the development of insulin resistance in obese liver and adipose tissue (*Yuan et al., 2001*; *Arkan et al., 2005*; *Wunderlich et al., 2008*; *Chiang et al., 2009*). NFκB is activated by the IκB kinase (IKK) family, which has four members: IKKα, IKKβ, IKKε, and TBK1. IKKα and IKKβ act with the scaffolding partner NEMO to activate NFκB (*Hacker and Karin, 2006*). Although pharmacologic inhibition or genetic ablation of IKKβ defined a role for this kinase in insulin resistance (*Yuan et al., 2001*; *Arkan et al., 2005*), the roles of the noncanonical kinases IKKε and TBK1 are less certain.

**eLife digest** Obesity is a complex metabolic disorder that is caused by increased food intake and decreased expenditure of energy. Obesity also increases the risk of developing type 2 diabetes, heart disease, stroke, arthritis, and certain cancers. There is considerable evidence to suggest that adipose tissue becomes less sensitive to catecholamines such as adrenaline in states of obesity, and that this reduced sensitivity in turn reduces energy expenditure. However, the details of this process are not fully understood.

It is well established that obesity generates a state of chronic, low-grade inflammation in liver and adipose tissue, accompanied by the secretion of signaling proteins that prevent fat cells from responding to insulin, which leads to type 2 diabetes. Activation of the NFκB pathway is thought to have a central role in causing this inflammation. Now Mowers et al. have investigated whether inflammation caused by activation of the NFκB pathway also has a role in producing catecholamine resistance in fat cells.

Obesity-dependent activation of the NFκB pathway increases the levels of a pair of enzymes, IKKε and TBK1. Mowers et al. found that elevated levels of these two enzymes reduced the ability of certain receptors (called β-adrenergic receptors) in the fat cells of obese mice to respond to catecholamines. High levels of the two enzymes also resulted in lower levels of a second messenger molecule called cAMP, which increases energy expenditure by elevating fat burning. However, treating the fat cells with drugs that interfere with the two enzymes restored sensitivity to catecholamine, allowing the fat cells to burn energy.

Mowers et al. also treated obese mice with amlexanox, a drug that inhibits these enzymes, and found that this treatment made the mice sensitive to a synthetic catecholamine that triggered the release of energy from fat. Mowers et al. suggest, therefore, that IKKε and TBK1 respond to inflammation in the body by reducing catecholamine signaling, thus preventing energy expenditure. Drugs targeting these enzymes may be useful for treating conditions like obesity or type 2 diabetes.

We recently reported that both mRNA and protein expression levels of IKKε and TBK1 are increased in adipose tissue from mice fed a high fat diet (*Chiang et al., 2009*). Both of these kinases are increased as a consequence of the inflammatory program in obesity (*Reilly et al., 2013*), and contain NFκB regulatory sites in their promoter regions, allowing them to be induced upon NFκB activation (*Kravchenko et al., 2003*). Deletion of the IKKε gene rendered mice partially resistant to some of the deleterious effects of high fat feeding, including weight gain, insulin resistance, hepatic steatosis, and systemic inflammation (*Chiang et al., 2009*). We report, in this study, that IKKε and TBK1 can desensitize lipolytic signaling in white adipose tissue in response to β-adrenergic agonists by phosphorylating and increasing the activity of PDE3B, in the process decreasing cAMP levels. Thus, induction of these noncanonical IκB kinases might contribute to catecholamine resistance during obesity, and blocking their activity has the potential to increase energy expenditure as an anti-obesity and anti-diabetes therapy.

## Results

### IKKε and TBK1 overexpression decrease sensitivity to the β-adrenergic/cAMP pathway in 3T3-L1 adipocytes

Sympathetic activation of adipose tissue plays a key role in maintaining energy balance by stimulating lipolysis and fat oxidation (*Coppack et al., 1994*; *Langin, 2006*; *Festuccia et al., 2011*). Activation of β-adrenergic signaling by either β-adrenergic agonists or cold exposure in white and brown adipose tissue initiates a cascade of events through cyclic AMP (cAMP), culminating in the transcriptional upregulation of Ucp1, which results in increased proton leak and energy expenditure (*Himms-Hagen et al., 2000*; *Cao et al., 2004*; *Yehuda-Shnaidman et al., 2010*). Our previous studies revealed that compared to wild-type (WT) controls, IKKε-deficient mice exhibited increased energy expenditure while on a high fat diet (HFD), accompanied by increased expression of Ucp1 in white adipose depots (*Chiang et al., 2009*). Interestingly, increased energy expenditure in IKKε-deficient mice was only seen in HFD-fed mice (*Chiang et al., 2009*), suggesting that upon induction of IKKε during obesity, the

kinase might repress an increased adaptive thermogenic response to overnutrition. To explore this possibility, we overexpressed IKKε in 3T3-L1 adipocytes and examined *Ucp1* gene expression after treatment with the non-selective β-adrenergic agonist, isoproterenol (ISO), or the β₃-adrenergic agonist, CL-316,243. Fold difference in *Ucp1* gene expression was calculated by normalization of relative *Ucp1* mRNA levels in treated relative to control samples. Treatment of empty vector-expressing cells with ISO or CL-316,243 resulted in a 1.6-fold or twofold increase in *Ucp1* mRNA levels, respectively (**Figure 1A**).

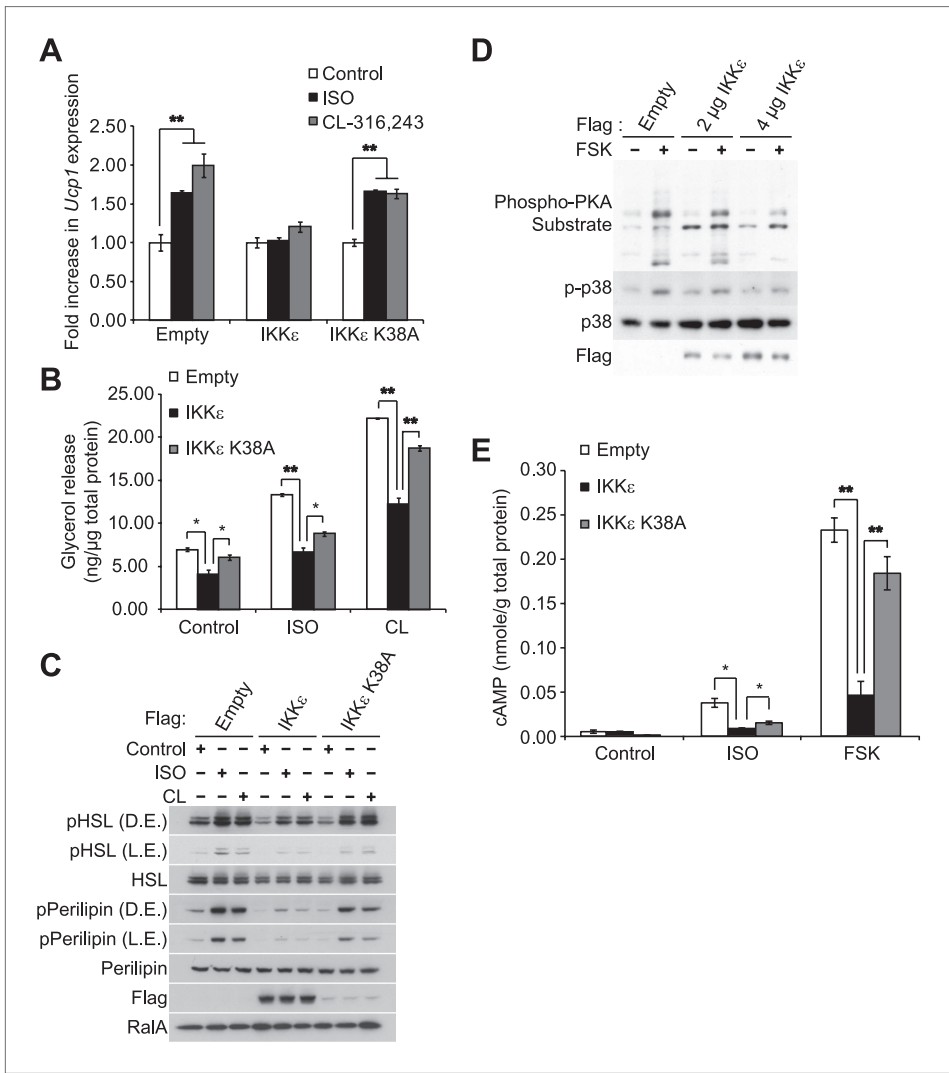

**Figure 1**. IKKε and TBK1 overexpression decrease sensitivity to the β-adrenergic/cAMP pathway in 3T3-L1 adipocytes. (**A**) Fold increase in *Ucp1* expression in 3T3-L1 adipocytes expressing empty vector, Flag-IKKε, or Flag-IKKε K38A following treatment with or without 10 μM ISO (black bars) or 10 μM CL-316,243 (CL, gray bars) for 4 hr. **p<0.01. Performed in triplicate. (**B**) Glycerol release from 3T3-L1 adipocytes expressing empty vector (white bars), Flag-IKKε (black bars), or Flag-IKKε K38A (gray bars) treated with or without 10 μM ISO or 10 μM CL. *p<0.05 and **p<0.01. Performed in triplicate. (**C**) Immunoblots of whole cell lysates from **Figure 1B**. Results were replicated in triplicate. D.E. stands for dark exposure and L.E. stands for light exposure. (**D**) Immunoblots of whole cell lysates from 3T3-L1 adipocytes expressing empty vector or Flag-IKKε treated with or without 50 μM FSK for 15 min. Results were replicated in multiple experiments. (**E**) cAMP levels from 3T3-L1 adipocytes expressing empty vector, Flag-IKKε, or Flag-IKKε K38A treated with or without 10 μM ISO or 50 μM FSK for 15 min. **p<0.0001 and *p<0.05. Performed in triplicate.

The following figure supplements are available for figure 1:

**Figure supplement 1**. IKKε and TBK1 overexpression decrease sensitivity to the β-adrenergic/cAMP pathway in 3T3-L1 adipocytes.

The induction of *Ucp1* gene expression in response to ISO or CL-316,243 was blunted when WT IKKε was overexpressed in these cells. However, expression of the kinase-inactive mutant of IKKε K38A (*Fitzgerald et al., 2003*) was less effective, but still modestly repressed *Ucp1* expression.

In addition to increased *Ucp1* expression, IKKε knockout mice also exhibited increased lipolysis and fat oxidation (*Chiang et al., 2009*), suggesting that decreased lipolysis in adipose tissue from obese mice might result in part from increased expression of IKKε and TBK1 (*Chiang et al., 2009*). We thus modeled the obesity-dependent increase in the noncanonical IKKs by overexpressing IKKε in 3T3-L1 adipocytes, followed by assay of glycerol release in response to ISO or CL-316,243. Although both isoproterenol and CL-316,243 increased lipolysis in empty vector-expressing cells, overexpression of WT IKKε reduced the lipolytic effects of isoproterenol and CL-316,243 by greater than 40%, and also reduced basal glycerol release (*Figure 1B*). The reduction in lipolysis by IKKε overexpression was accompanied by dramatically reduced phosphorylation of HSL and perilipin in response to ISO or CL-316,243 (*Figure 1C*). Expression of the catalytically inactive kinase was less effective in blocking lipolytic signaling, although the levels of protein achieved by overexpression were lower compared to the WT kinase (*Figure 1B,C*, *Figure 1—figure supplement 1A*). Overexpression of TBK1 reduced phosphorylation of HSL in response to isoproterenol or the adenylyl cyclase activator, forskolin (*Figure 1—figure supplement 1B*). Identical results were obtained when IKKε was overexpressed in 3T3-L1 adipocytes stimulated with forskolin (*Figure 1D*), as detected by western blotting with an anti-phospho-PKA substrate motif antibody. Overexpression of IKKε also repressed the phosphorylation of p38 (p-p38) in response to forskolin (*Figure 1D*) or isoproterenol (*Figure 1—figure supplement 1A*), whereas overexpression of IKKε K38A was without effect (*Figure 1—figure supplement 1A*). While glycerol release is likely the result of changes in HSL and perilipin phosphorylation, it is important to note that we have not directly assayed whether re-esterification of glycerol intermediates are also affected. Taken together, these data suggest that similar to what is observed in obesity, overexpression of IKKε or TBK1 can repress lipolytic signaling. The partial effectiveness of the kinase-inactive mutants is puzzling, but may reflect their activation of endogenous IKKε or TBK1 kinases due to dimerization (*Larabi et al., 2013*; *Tu et al., 2013*).

Since PKA signaling is responsible for Ucp1 induction in response to catecholamines (*Klein et al., 2000*; *Cao et al., 2001*), we explored the possibility that both IKKε and TBK1 might reduce β-adrenergic sensitivity of adipocytes by decreasing cAMP levels. IKKε overexpression in 3T3-L1 adipocytes reduced by greater than 80% the increase in cAMP levels produced by both isoproterenol and forskolin, whereas overexpression of IKKε K38A did not (*Figure 1E*). Previous studies have shown that decreased sensitivity to adrenergic stimuli in adipose tissue can result from reduced β-adrenergic receptors (*Reynisdottir et al., 1994*) or increased expression of α2-adrenergic receptors (*Stich et al., 2002*). These studies represent the first demonstration that defects distal to the adrenergic receptor may also contribute to catecholamine resistance, and suggest that IKKε and TBK1 can attenuate the β-adrenergic/cAMP pathway in response to β-adrenergic stimuli in adipocytes in a cell-autonomous manner, and further that induction of these kinases during obesity may account for decreased energy expenditure by reducing sensitivity of adipocytes to β-adrenergic stimulation.

## Prolonged treatment with TNFα decreases the sensitivity of adipocytes to β-adrenergic stimulation in a manner dependent on the activity of IKKε and TBK1

Obesity is accompanied by infiltration of proinflammatory macrophages into adipose tissue; these cells secrete inflammatory cytokines, such as TNFα, which generate insulin resistance by stimulating catabolic pathways (*Hotamisligil, 2006*; *Lumeng et al., 2007*; *Ye and Keller, 2010*; *Ouchi et al., 2011*). Although TNFα is known to increase lipolysis in adipocytes (*Zhang et al., 2002*; *Souza et al., 2003*; *Green et al., 2004*; *Plomgaard et al., 2008*), there is also evidence of a counterinflammatory response in obesity that may serve to repress energy expenditure (*Gregor and Hotamisligil, 2011*; *Saltiel, 2012*; *Calay and Hotamisligil, 2013*; *Reilly et al., 2013*). We thus used TNFα to model the inflammatory milieu of obese adipose tissue in cell culture to determine whether the cytokine might also regulate β-adrenergic signaling in this context. While short-term treatment with TNFα augmented the increase in cAMP produced by forskolin treatment, this effect declined after 12 hr. After 24 hr of exposure, TNFα inhibited the production of the second messenger produced by forskolin (*Figure 2—figure supplement 1A*). Thus, the catabolic effects of the proinflammatory cytokine TNFα in adipocytes are transient and followed by an inhibitory phase.

Our previous studies revealed that treatment of 3T3-L1 adipocytes with TNFα for 24 hr induced the expression of IKKε and increased TBK1 phosphorylation at the active site in a manner that was dependent on the activity of IKKβ and the NFκB pathway (*Reilly et al., 2013*). We thus wondered whether the repression of β-adrenergic sensitivity produced by longer-term treatment with TNFα might be due to increased activity of the noncanonical IKKs. Long-term treatment with TNFα repressed the induction of *Ucp1* gene expression in response to β-adrenergic stimuli (*Figure 2—figure supplement 1B*), whereas the expression of IKKε mRNA (*Ikbke*) was upregulated, as previously reported (*Reilly et al., 2013*). Treatment of 3T3-L1 adipocytes with TNFα for 24 hr decreased glycerol release in response to both isoproterenol and forskolin in a dose-dependent manner (*Figure 2A*). TNFα treatment also decreased isoproterenol- and forskolin-stimulated cAMP production; an effect that was largely rescued by preincubation of cells with the selective, but structurally unrelated inhibitors of IKKε and TBK1, amlexanox (*Figure 2B*) (*Reilly et al., 2013*) or CAY10576 (*Figure 2C*) (*Bamborough et al., 2006*).

Isoproterenol-stimulated β-adrenergic signaling was also decreased by treatment of cells with TNFα (*Figure 2D*), as manifested by decreased phosphorylation of HSL, perilipin, and other proteins recognized by the PKA substrate motif antibody, whereas IKKε expression was concurrently upregulated and TBK1 phosphorylation was increased by the treatment with TNFα. Pretreatment of 3T3-L1 adipocytes with amlexanox also blocked the inhibitory effect of TNFα on isoproterenol-stimulated β-adrenergic signaling, as determined by western blotting with an anti-phospho-PKA substrate motif antibody, anti-phospho-HSL, and anti-phospho-perilipin antibodies (*Figure 2E*). Interestingly, phosphorylation of p38 in response to isoproterenol was also dramatically augmented by amlexanox in a dose-dependent manner. Previous studies showed that the toll-like receptor 3 (TLR3) agonist, Poly (I:C), results in the direct activation of IKKε and TBK1 (*Hemmi et al., 2004*; *Clark et al., 2009*; *Clark et al., 2011*). Similar to TNFα, treatment of 3T3-L1 adipocytes with Poly (I:C) simultaneously reduced stimulation of cAMP production, lipolysis and phosphorylation in response to β-adrenergic stimulation (*Figure 2—figure supplement 1C–E*), and the inhibitory effects of Poly (I:C) on the sensitivity to isoproterenol stimulation were partially restored by amlexanox pretreatment, but not to the extent that was observed with TNFα treatment (*Figure 2E*). It is possible that Poly (I:C)-induced desensitization of β-adrenergic pathway engages other pathways that are not directly regulated by IKKε and TBK1. These results suggest that obesity-associated inflammation leads to the activation of IKKε and TBK1, which produces reduced sensitivity of adipocytes to β-adrenergic stimulation.

## IKKε and TBK1 reduce cAMP levels through activation of PDE3B

cAMP levels can also be regulated by phosphodiesterases, which cleave the second messenger and in the process dampen cAMP-dependent signals. Phosphodiesterase 3B (PDE3B) is the major PDE isoform expressed in adipocytes (*Zmuda-Trzebiatowska et al., 2006*). Genetic ablation or pharmacological inhibition of PDE3B in cells and *in vivo* revealed an important role for the enzyme in lipid and glucose metabolism (*Choi et al., 2006*; *Berger et al., 2009*; *Degerman et al., 2011*). Phosphorylation and activation of PDE3B by insulin in adipocytes is thought to be mediated by Akt, and cAMP itself acts as a negative feedback regulator of its own levels by promoting PKA-dependent phosphorylation and activation of PDE3B (*Degerman et al., 2011*).

Since we observed that cAMP production was impaired in forskolin or isoproterenol-stimulated 3T3-L1 adipocytes overexpressing IKKε (*Figure 1E*), we examined whether noncanonical IKKs might desensitize adrenergic stimulation through increased activity of PDE3B in adipocytes. Pretreatment with a nonspecific phosphodiesterase inhibitor, IBMX, in 3T3-L1 adipocytes expressing IKKε or TBK1 rescued the full stimulation of cAMP production in response to forskolin (*Figure 3A*). Interestingly, the selective PDE3B and PDE4 inhibitor, Zardaverine (*Schudt et al., 1991*), also blocked the inhibitory effects of IKKε and TBK1 overexpression on cAMP levels in response to isoproterenol and forskolin in 3T3-L1 adipocytes (*Figure 3B*), suggesting an important role for PDE3B as a target of the noncanonical IKKs.

We next examined whether IKKε and TBK1 directly phosphorylate PDE3B to regulate cAMP levels. Recombinant TBK1, Akt and PKA were incubated *in vitro* with [γ-$^{32}$P]ATP and purified PDE3B as a substrate. Phosphorylation was assessed by SDS-PAGE followed by autoradiography. TBK1 directly catalyzed the phosphorylation of PDE3B; phosphorylation was also produced by incubation with Akt and PKA, as previously reported (*Kitamura et al., 1999*; *Palmer et al., 2007*) (*Figure 3C*). IKKε also catalyzed this phosphorylation *in vitro* (data not shown). This increase in phosphorylation produced by *in vitro* incubation with TBK1, IKKε and PKA was also detected when PDE3B was blotted with antibodies

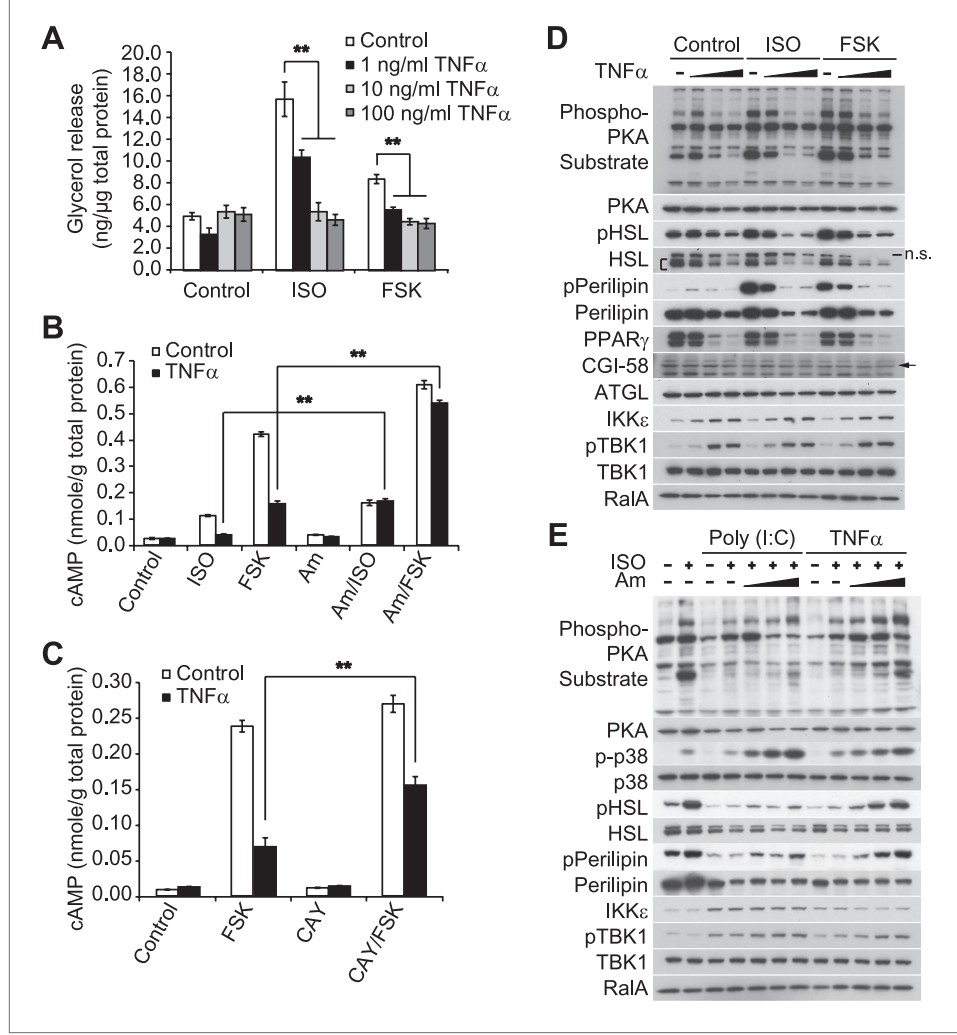

**Figure 2**. Prolonged treatment with TNFα decreases the sensitivity of adipocytes to β-adrenergic stimulation in a manner dependent on the activity of IKKε and TBK1. (**A**) Glycerol release from 3T3-L1 adipocytes treated with or without different concentrations of TNFα as indicated for 24 hr followed by treatment with or without 10 µM ISO or 50 µM FSK. **p<0.0001. Performed in quadruplicate. (**B**) cAMP levels from 3T3-L1 adipocytes treated with or without 100 ng/ml TNFα for 24 hr followed by treatment with or without 10 µM ISO or 50 µM FSK in the presence or absence of pretreatment of 50 µM Amlexanox (Am). **p<0.0001. Performed in quadruplicate. (**C**) cAMP levels from 3T3-L1 adipocytes treated with or without 100 ng/ml TNFα for 24 hr followed by treatment with or without 50 µM FSK in the presence or absence of pretreatment of 1 µM CAY10576 (CAY). **p<0.0001. Performed in triplicate. (**D**) Immunoblots of whole cell lysates from 3T3-L1 adipocytes treated with or without different concentrations of TNFα as same as *Figure 2A* for 24 hr followed by treatment with or without 10 µM ISO or 50 µM FSK. Results were replicated in multiple experiments. '[' indicates total HSL. 'n.s.' represents non-specific band. Arrow indicates CGI-58. (**E**) Immunoblots of whole cell lysates from 3T3-L1 adipocytes treated with or without 50 ng/ml TNFα or 100 µg/ml poly (I:C) for 24 hr followed by treatment with or without 10 µM ISO for 15 min in the presence or absence of pretreatment with increasing concentrations (0, 10, 50, and 200 µM) of amlexanox for 30 min. Results were replicated in multiple experiments.

The following figure supplements are available for figure 2:

**Figure supplement 1**. Prolonged exposure of inflammatory cytokines decreases the sensitivity of adipocytes to β-adrenergic stimulation.

that recognize the 14-3-3 binding motif (*Figure 3—figure supplement 1A*). When purified PDE3B was incubated with the same amount of recombinant TBK1 and canonical IKKβ kinases *in vitro*, phosphorylation of PDE3B by IKKβ was barely detectable, indicating a level of specificity in which PDE3B is

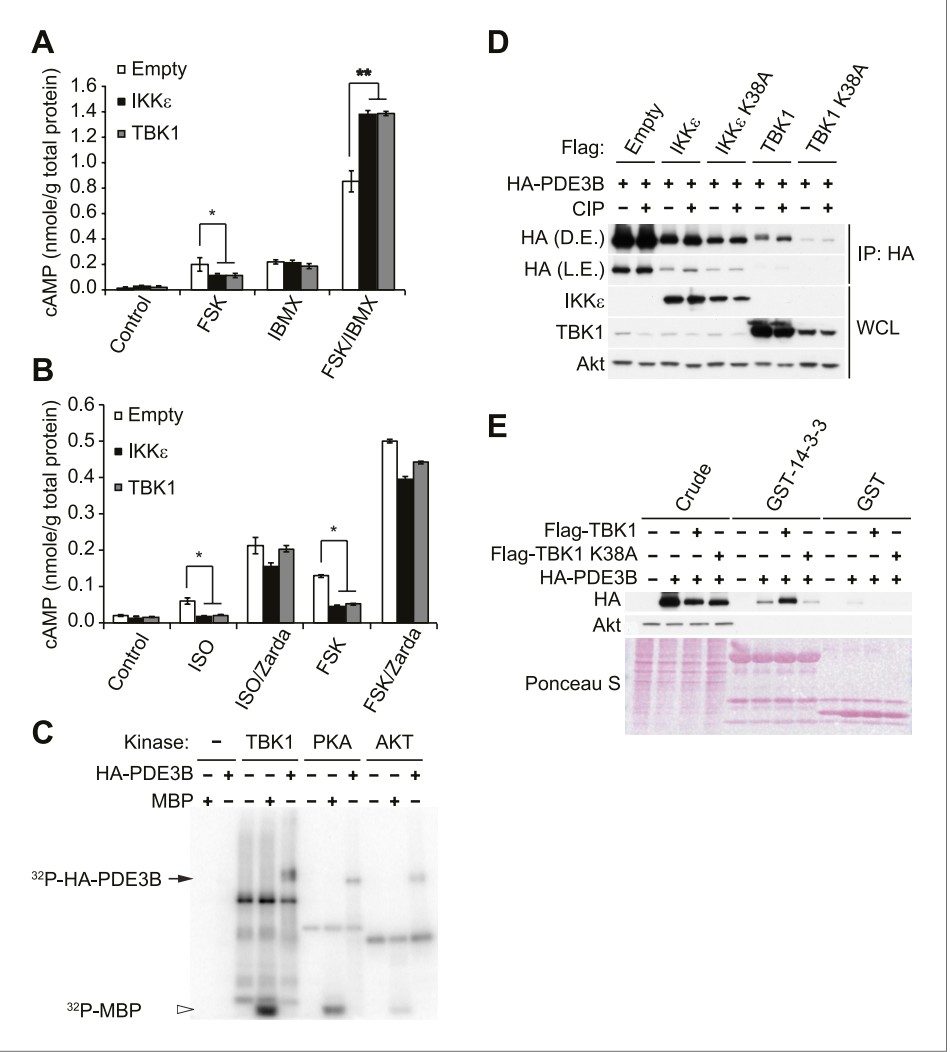

**Figure 3**. IKKε and TBK1 reduce cAMP levels through activation of PDE3B. (**A**) cAMP levels from 3T3-L1 adipocytes expressing empty vector, Flag-IKKε, or Flag-TBK1 treated with or without 50 μM FSK, 250 μM IBMX, or together for 15 min. *p<0.05 and **p<0.0001. Performed in duplicate. (**B**) cAMP levels from 3T3-L1 adipocytes expressing empty vector, Flag-IKKε, or Flag-TBK1 treated with or without 10 μM ISO or 50 μM FSK together with or without 10 μM Zardaverine (Zarda) for 15 min. *p<0.05. Performed in duplicate. (**C**) $^{32}$P phospho-image of *in vitro* kinase reaction using either immunoprecipitated HA-PDE3B from HEK293T cells or 1 μg MBP (myelin basic protein) as a substrate with recombinant kinases as indicated. Results were replicated in multiple experiments. (**D**) Immunoblots of immunoprecipitation with anti-HA antibodies followed by treatment with or without CIP (top panel) and whole cell lysates (bottom panel) from Cos-1 cells co-expressing HA-PDE3B with Flag-IKKε/TBK1 or Flag-IKKε/TBK1 K38A. D.E. stands for dark exposure and L.E. stands for light exposure. Results were replicated in multiple experiments. (**E**) Immunoblots of GST-14-3-3 pulldown from HEK293T cells co-expressing HA-PDE3B with Flag-TBK1 or Flag-TBK1 K38A. Ponceau S staining shows the amount of beads used in GST-14-3-3 pulldown. Results were replicated in multiple experiments.

The following figure supplements are available for figure 3:

**Figure supplement 1**. IKKε and TBK1 interact with PDE3B in a manner dependent on the activity of IKKε and TBK1.

a better target of the noncanonical IKKs (***Figure 3—figure supplement 1B***). This phosphorylation was dose-dependent with respect to ATP (***Figure 3—figure supplement 1C***).

To determine whether IKKε can phosphorylate PDE3B in cells, we co-expressed IKKε and its inactive mutant K38A with HA-tagged PDE3B in HEK293T cells, followed by immunoprecipitation (IP) with anti-HA antibodies. Expression of IKKε in cells caused a shift in electrophoretic mobility of PDE3B, and

this shift was not detected when IKKε K38A was expressed (*Figure 3—figure supplement 1D*). Phosphorylation of PDE3B was also detected after expression of IKKε but not its kinase-inactive mutant K38A in cells, as detected by blotting with antibodies that recognize the 14-3-3 binding motif. To determine whether this molecular shift was dependent on phosphorylation of PDE3B, HA-PDE3B was co-expressed in Cos-1 cells along with IKKε, TBK1 or their kinase inactive mutants, and HA immuno-precipitates were treated with or without calf intestinal phosphatase (CIP). Expression of both of the wild-type kinases reduced the electrophoretic mobility of PDE3B, which could be reversed by treatment with the phosphatase (*Figure 3D*, compare lane 3, 7 to lane 4, 8). Neither of the kinase-inactive mutants had an effect (*Figure 3D*, compare lane 5, 9 to lane 6, 10).

Previous studies suggested that IKKε and TBK1 bind to their respective substrates through a sequence that includes a ubiquitin-like domain (ULD) proximal to their kinase domain. This domain is highly conserved among the IKK family members, and is 49% identical between IKKε and TBK1 (*Ikeda et al., 2007*; *May et al., 2004*). To confirm that PDE3B is a bona fide substrate of IKKε and TBK1, we prepared a GST-ULD domain fusion protein from TBK1 and incubated this fusion protein with 3T3-L1 adipocyte lysates. The fusion protein specifically precipitated endogenous PDE3B from these lysates (*Figure 3—figure supplement 1E*). To explore further the interaction of these two proteins, we co-expressed WT TBK1 and its K38A mutant with HA-tagged PDE3B in HEK293T cells, and immunoprecipitated the protein with anti-HA antibodies. Kinase-inactive TBK1 was preferentially co-immunoprecipitated with PDE3B, whereas the interaction of PDE3B with WT TBK1 was barely detectable (*Figure 3—figure supplement 1F*). These data suggest that TBK1 and IKKε associate with substrates such as PDE3B, and subsequently dissociate upon phosphorylation.

Next, to test further the role of PDE3B phosphorylation by IKKε and TBK1 in initiating its interaction with 14-3-3β, we prepared a GST-14-3-3β fusion protein which was incubated with lysates from HEK293T cells co-expressing TBK1 with PDE3B. PDE3B was preferentially pulled down by GST-14-3-3β after phosphorylation by TBK1 but not by its inactive K38A mutant, whereas GST beads alone enriched neither PDE3B nor its phosphorylated form (*Figure 3E*).

## IKKε and TBK1 phosphorylate PDE3B at serine 318, resulting in the binding of 14-3-3β

To evaluate the regulatory role of PDE3B phosphorylation by IKKε and TBK1, we determined which sites are phosphorylated. HA-PDE3B was co-expressed in Cos-1 cells with IKKε and TBK1, and phos-phorylated PDE3B was enriched by IP with anti-HA antibodies. Phosphorylation sites on human PDE3B were then determined by LC-MS/MS mass spectrometry. This analysis revealed that serines 22, 299, 318, 381, 463, 467, and 503 were phosphorylated by both kinases; there were no differences between the kinases (*Figure 4A*). Interestingly, the phosphorylation profile of PDE3B matched neither known Akt or PKA profiles (*Lindh et al., 2007*). However, phosphorylation on serine 299 and serine 318 had previously been identified on mouse PDE3B (residues equivalent to Serine 277 and 296 in mouse PDE3B) in adipocytes and hepatocytes in response to both insulin and forskolin (*Lindh et al., 2007*).

While several serine residues are known to be phosphorylated on PDE3B in response to stimuli, serine 318 (human) is the best characterized. This residue resides in a consensus phosphorylation sequence for both Akt and PKA, and also serves as a consensus 14-3-3 binding motif once phosphorylated (*Lindh et al., 2007*; *Palmer et al., 2007*). We thus created a Ser318Ala (S318A) mutant of PDE3B, and examined its interaction with a GST-14-3-3β fusion protein or by GST-14-3-3 overlay assay. Interestingly, despite incubation with TBK1, the phospho-defective, S318A mutant of PDE3B, did not specifically interact with GST-14-3-3β, whereas the wild-type protein did (*Figure 4B,C*). In a GST pull-down assay, the molecular shift of PDE3B S318A was still detected by western blot (*Figure 4B*), indicating that phosphorylation of PDE3B by TBK1 on other sites still occurred, but were not crucial for 14-3-3β binding.

To examine the functional importance of the phosphorylation of PDE3B at Serine 318, we overex-pressed WT PDE3B and its S318A mutant in 3T3-L1 adipocytes, and tested the response of the cells to TNFα. Overexpression of WT PDE3B in cells reduced the attenuation of forskolin-stimulated cAMP production and phosphorylation of HSL produced by TNFα, whereas PDE3B S318A was ineffective (*Figure 4D*, *Figure 4—figure supplement 1A,B*). These data suggest that although IKKε and TBK1 can phosphorylate PDE3B on several sites, serine 318 may be particularly important in the regulation of phosphodiesterase function by promoting the interaction between PDE3B and 14-3-3β. More importantly, this residue is the major site mediating the negative effects of IKKε and TBK1 on sensitivity of adipocytes to β-adrenergic stimulation.

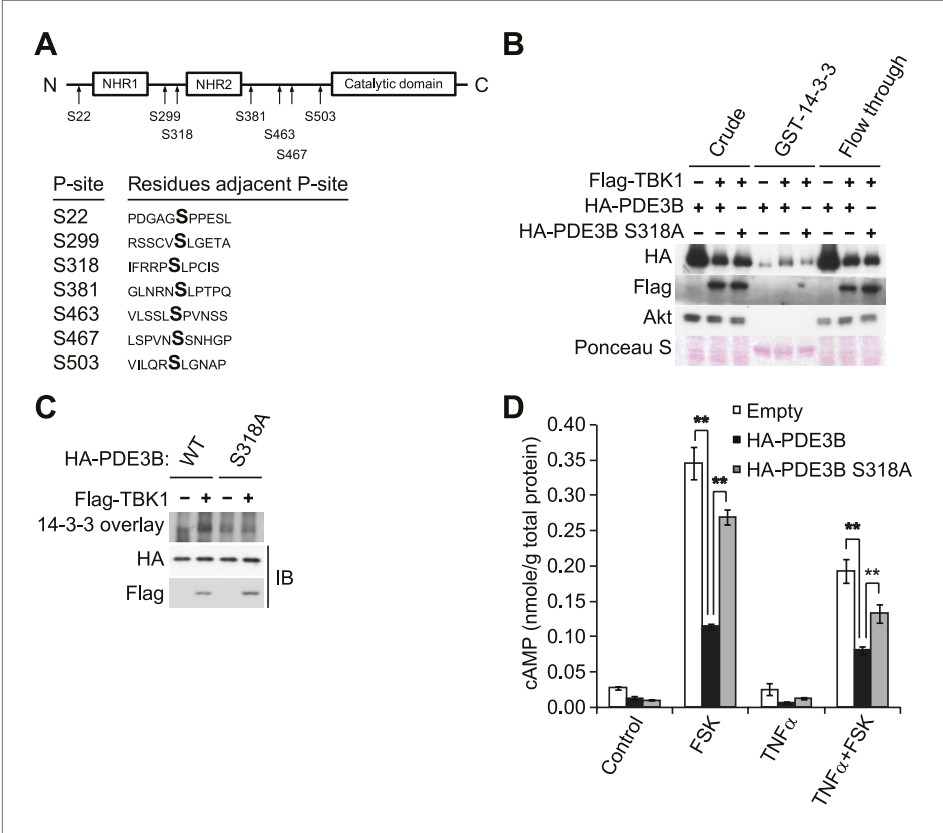

**Figure 4**. IKKε and TBK1 phosphorylate PDE3B at serine 318, resulting in the binding of 14-3-3β. (**A**) Summary of sites on PDE3B phosphorylated by IKKε or TBK1 (P-sites) from mass spectrometry experiments. (**B**) Immunoblots of GST-14-3-3 pulldown from HEK293T cells co-expressing HA-PDE3B or HA-PDE3B S318A with Flag-TBK1. Ponceau S staining shows the amount of beads used in GST-14-3-3 pulldown. Results were replicated in multiple experiments. (**C**) GST-14-3-3 overlay on nitrocellulose membrane (top blot) and an immunoblot (IB) of whole cell lysates from HEK293T cells co-expressing HA-PDE3B or HA-PDE3B S318A with Flag-TBK1 (bottom blot). Results were replicated in multiple experiments. (**D**) cAMP levels from 3T3-L1 adipocytes expressing empty vector, HA-PDE3B, or HA-PDE3B S318A treated with or without 100 ng/ml TNFα for 16 hr followed by treatment with or without 25 μM FSK for 15 min. **p<0.0001 and **p<0.01. Performed in duplicate.

The following figure supplements are available for figure 4:

**Figure supplement 1**. Overexpression of PDE3B in 3T3-L1 adipocytes reduces the attenuation of forskolin-stimulated β-adrenergic signaling produced by TNFα.

## The IKKε/TBK1 inhibitor Amlexanox sensitizes β-adrenergic agonist-stimulated lipolysis in white adipose tissue in diet-induced obese mice

Disruption of sympathetic activation of lipolysis and fat oxidation may play an important role in the development and maintenance of increased fat storage in obesity. Indeed, while numerous studies have demonstrated catecholamine resistance in obese adipose tissue (*Jensen et al., 1989*; *Reynisdottir et al., 1994*; *Bougneres et al., 1997*; *Arner, 1999*; *Jocken and Blaak, 2008*), the underlying mechanisms remain unclear. To test the functional importance of the noncanonical IKKs in maintaining energy balance *in vivo*, we investigated whether the administration of a selective inhibitor of IKKε and TBK1, amlexanox, can reverse diet-induced catecholamine resistance in rodents. We fed mice a high fat or normal diet, treated them with amlexanox by oral gavage for 4 days (prior to the point when weight loss is seen), and then gave a single intraperitoneal (IP) injection of the β₃-adrenergic agonist CL-316,243. Injection of CL-316,243 stimulated a threefold increase in serum FFA and glycerol levels in both vehicle and amlexanox-treated mice on normal diet (ND). The effect of CL-316,243 to increase serum FFAs was significantly attenuated in HFD-fed, vehicle-treated mice. However, HFD-fed mice treated with

amlexanox responded like normal diet mice, despite the fact that they were weight matched with control HFD-fed mice (*Figure 5A*). The fold increase in serum glycerol levels was also significantly higher in amlexanox-treated HFD mice, as compared to vehicle-treated HFD-fed mice. In addition, *ex vivo* pretreatment of white adipose tissues from mice on a HFD with amlexanox enhanced glycerol release (*Figure 5B*). This effect was more pronounced in the inguinal fat depot, where amlexanox pretreatment increased phosphorylation of HSL, perilipin, and other proteins recognized by the PKA substrate motif antibody in response to CL-316,243 treatment compared to vehicle-pretreated tissues (*Figure 5C*). Amlexanox also concurrently increased the phosphorylation of TBK1 at Ser172 due to the relief of feed-back inhibition, as previously reported with other inhibitors (*Clark et al., 2009*; *Reilly et al., 2013*).

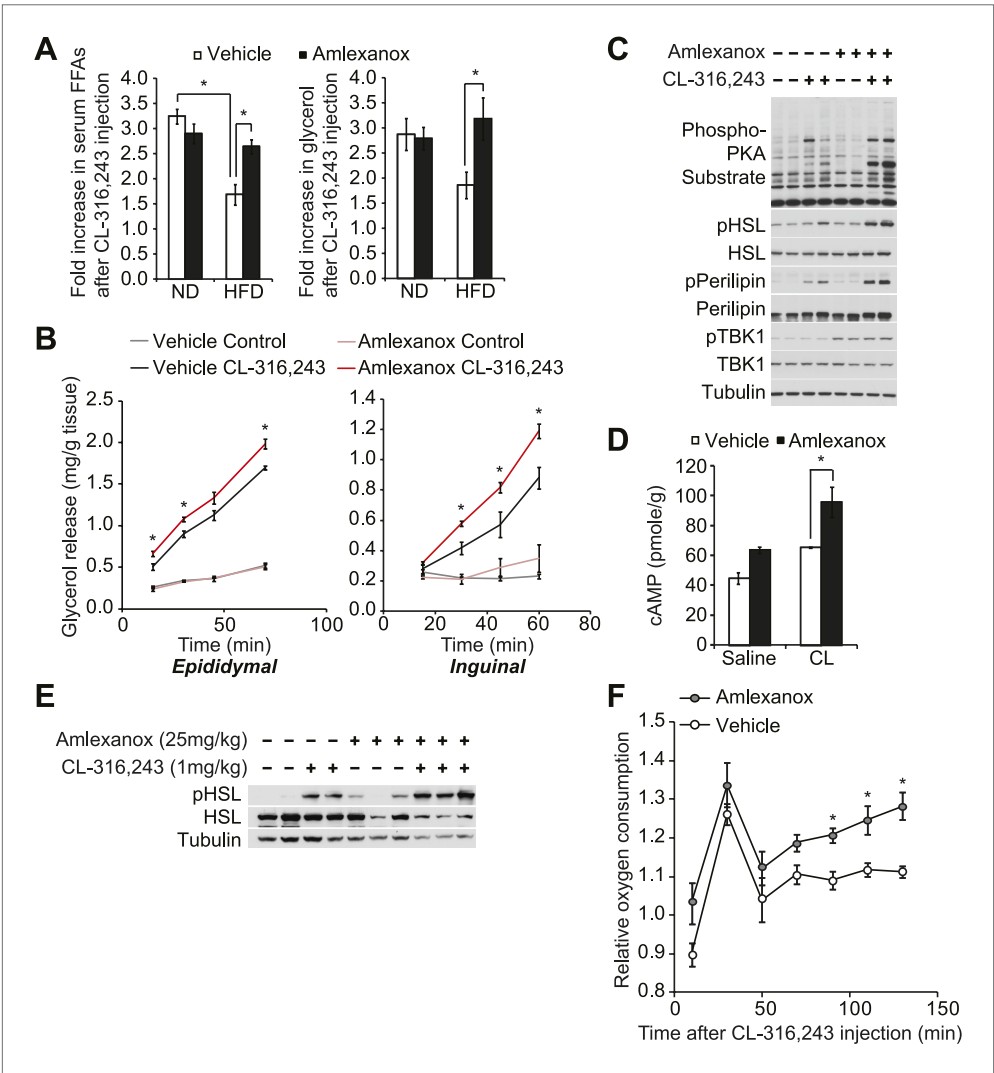

**Figure 5**. The IKKε/TBK1 inhibitor Amlexanox sensitizes β-adrenergic agonist-stimulated lipolysis in white adipose tissue in diet-induced obese mice. (**A**) Fold increase in serum FFA (left panel) and glycerol (right panel) levels 15 min after CL-316,243 injection in ND- or HFD-fed mice treated with amlexanox or vehicle control for 4 days. n = 7 mice per group. *p<0.05. (**B**) Glycerol release from *ex vivo* epididymal (left panel) and inguinal (right panel) WATs after 1 hr pretreatment with amlexanox or vehicle. CL-316,243 treatment was started at time zero. n = 6, 3 WAT pieces × 2 mice. *P<0.05. (**C**) Immunoblots in inguinal WAT lysates from *Figure 5B* after 60 min of CL-316,243 treatment. (**D**) cAMP levels in epididymal WAT 20 min after CL-316,243 (CL) or saline control injection in HFD-fed mice treated with amlexanox or vehicle control for 4 days. n = 2 mice per saline-treated group and n = 3 mice per CL-316,324-treated group. *p<0.05. (**E**) Immunoblots in epididymal WAT 5 min after CL-316,243 or saline control injection in HFD-fed mice treated with amlexanox or vehicle control for 4 days. (**F**) Relative oxygen comsumption of mice in each treatment group. n = 7 for the vehicle-treated group, n = 5 for the amlexanox-treated group. *P<0.05 (Student's *t* test).

To examine whether inhibition of TBK1 and IKKε with amlexanox reverses resistance to catecholamine-induced lipolysis *in vivo* by increasing stimulation of cAMP production, we measured cAMP levels in epididymal adipose tissue from mice on HFD after CL-316,243 IP injection. Interestingly, levels of cAMP were increased after CL-316,243 IP injection in mice on HFD pretreated with amlexanox (*Figure 5D*). Consistent with this, HSL phosphorylation was also increased after CL-316,243 IP injection of HFD-fed mice pretreated with amlexanox (*Figure 5E*).

Our previous studies showed that increased expression of Ucp1 in white adipose depots resulted in increased energy expenditure in IKKε-deficient mice (*Chiang et al., 2009*) and amlexanox-treated mice (*Reilly et al., 2013*) while on a high fat diet but not on a normal diet. To examine whether inhibition of catecholamine resistance in obese adipose tissue by targeting noncanonical IKKs with amlexanox can lead to increase energy expenditure in diet-induced obese mice, we measured oxygen consumption rates of vehicle or amlexanox-treated HFD-fed mice after a single injection of CL-316,243 in metabolic cages. The effect of CL-316,243 to increase energy expenditure was more pronounced in amlexanox-treated HFD-fed mice, as compared to vehicle-treated HFD-fed mice (*Figure 5F*). These data suggest that targeting the noncanonical IKKs with the selective inhibitor amlexanox ameliorated catecholamine resistance in obese adipose tissue.

## Discussion

Decreased sympathetic activation of adipose tissue due to impaired catecholamine synthesis or sensitivity has been observed in obese patients (*Reynisdottir et al., 1994*; *Stallknecht et al., 1997*; *Horowitz and Klein, 2000*; *Jocken et al., 2008*). Obesity is commonly associated with blunted whole-body catecholamine-induced lipolysis (*Horowitz and Klein, 2000*). This is thought to occur through a number of mechanisms, including leptin resistance (*Myers et al., 2010*), as well as the reduced expression of β-adrenergic receptors (*Reynisdottir et al., 1994*) or increased expression of $\alpha_2$-adrenergic receptors (*Stich et al., 2002*). White adipose tissue and cultured isolated adipocytes from obese human and mouse models exhibit decreased cAMP-stimulated lipolysis and fat oxidation, due to reduced energy expenditure from decreased mitochondrial uncoupling (*Yehuda-Shnaidman et al., 2010*). This desensitization to adrenergic activation is also a feature of childhood onset obesity (*Bougneres et al., 1997*; *Enoksson et al., 2000*), and has been observed in adipocytes from first-degree relatives of obese subjects (*Hellstrom et al., 1996*).

We demonstrate here a novel link between obesity and reduced sympathetic activity and β-adrenergic sensitivity, through the inflammation-dependent induction of the noncanonical IκB kinases IKKε and TBK1. Obesity generates a state of low-grade inflammation in both humans and rodents, which involves activation of the NFκB pathway (*Wellen and Hotamisligil, 2005*; *Hotamisligil, 2006*; *Shoelson et al., 2007*). Upon its prolonged activation, NFκB induces the expression of the noncanonical IκB kinases IKKε and TBK1. The induction of these kinases was blocked by administration of anti-inflammatory agents to mice without producing weight loss, suggesting that they are expressed in response to inflammation rather than obesity *per se* (*Reilly et al., 2013*). Deletion of the IKKε gene rendered mice partially resistant to weight gain, insulin resistance, steatosis and the long-term inflammation produced by high fat diet (*Chiang et al., 2009*), and administration of the dual specificity IKKε/TBK1 inhibitor amlexanox to diet-induced obese or ob/ob mice produced even more profound effects (*Reilly et al., 2013*). The blockade of these kinases in obese rodents with amlexanox results in increased phosphorylation of PKA substrates in adipose tissue, along with increased expression of Ucp1, and improved rates of lipolysis and fat oxidation (*Reilly et al., 2013*). Amlexanox was shown to inhibit phosphodiesterase activity of rat peritoneal mast cells via an unknown mechanism (*Makino et al., 1987*). Together these data indicate that IKKε and TBK1 might exert their physiological effects in part by reducing the sensitivity of adipocytes to β-adrenergic stimulation via changes in cAMP.

Data presented here suggest that the molecular target of IKKε/TBK1 is the phosphodiesterase PDE3B. Upon increased expression in the obese state, these kinases can phosphorylate PDE3B, causing an increase in the activity of the enzyme that cleaves cAMP, reducing the stimulation of cAMP-dependent phosphorylation of proteins in response to sympathetic activation. These proteins include HSL and perilipin, responsible for β-adrenergic-stimulated lipolysis, and other proteins such as p38 that regulate expression of Ucp1. The reduced sensitivity to β-adrenergic activation can attenuate lipolysis and fatty acid oxidation, as well as adaptive thermogenesis.

Several issues deserve further attention. The first concerns the relative roles of the two noncanonical IKKs in this pathway. Both TBK1 and IKKε are induced in response to obesity-dependent inflammation,

and appear to phosphorylate PDE3B on the same residues with equal efficiency. Although there are differences in expression of these kinases in other tissues (*Shimada et al., 1999*), and perhaps differences in the upstream signals that lead to their regulation (*Wunderlich et al., 2008*), their relative roles in controlling this pathway remain uncertain. Additionally, the mechanism by which PDE3B is regulated remains uncertain. While phosphorylation correlates well with decreased levels of cAMP in cells, we have been unable to demonstrate increased catalytic activity of the enzyme due to phosphorylation by IKKε, TBK1 or the other kinases (*Kitamura et al., 1999*; *Palmer et al., 2007*) thought to regulate the phosphodiesterase. Perhaps the phosphorylation-dependent binding of the enzyme to 14-3-3 exerts changes in its localization and access to its substrate, thus explaining increased activity in cells.

How is it that insulin resistance produced by inflammation fails to block continued energy storage? One possible explanation may lie in the homeostatic response to inflammation itself, typified by the induction of TBK1 and IKKε. Our data confirm previous findings that TNFα and perhaps other inflammatory cytokines can promote lipolytic processes in cells after short-term treatment, but that after longer exposure elicit an inhibitory response that appears to be the result of TBK1 and IKKε induction. Thus, these kinases may be part of a 'counter-inflammatory' program that attenuates the extent to which inflammatory signals are effective, and also serves to conserve energy by repressing lipolysis and fatty acid oxidation through activation of PDE3B. Interestingly, PDE3B is also a target of insulin action in adipocytes (*Degerman et al., 2011*). Thus, TBK1 and IKKε appear to co-opt insulin targets to conserve energy during obesity. These insights further suggest that the noncanonical IKKs might be interesting new therapeutic targets for the treatment of obesity and type 2 diabetes.

## Materials and methods

### Reagents

All chemicals were obtained from Sigma-Aldrich (Saint Louis, MO) unless stated otherwise. Anti-Flag antibody was obtained from Sigma, and anti-HA antibody was obtained from Santa Cruz Biotechnology (Santa Cruz, CA). Anti-IKKε, anti-TBK1, anti-phospho-TBK1 (Ser172), anti-AKT, anti-phospho-AKT (Ser473), anti-HSL, anti-phospho-HSL (Ser660), anti-p38, anti-phospho-p38, anti-perilipin, anti-ATGL and anti-PPARγ antibodies were purchased from Cell Signaling Technology (Danvers, MA). Anti-phospho-perilipin (Ser522) was purchased from Vala Sciences Inc (San Diego, CA). Anti-CGI-58 was purchased from Novus Biologicals (Littleton, CO). Anti-RalA antibody was obtained from BD Bioscience (San Jose, CA). Anti-Ucp1 antibody was obtained from Alpha Diagnostics (San Antonio, TX). Anti-PDE3B was provided as a generous gift by the Dr Vince Manganiello (NHLBI, NIH). Enhanced chemiluminescence (ECL) reagents were purchased from Thermo Scientific (Rockford, IL). EDTA-free protease inhibitor tablet was purchased from Roche Diagnostics (Indianapolis, IN). Monoclonal anti-HA agarose (Sigma) was used for immunoprecipitations, performed using the manufacturer's protocol. The human PDE3B cDNA was kindly provided by Dr Morris Birnbaum (University of Pennsylvania). The human 14-3-3β cDNA was kindly provided by Dr Ken Inoki (University of Michigan) and subcloned into pKH3 (*Chen et al., 2007*) and pGEX-4T-1 vectors (GE Healthcare Life Sciences, MI). Amlexanox was purchased from Ontario Chemical Inc. (Guelph, Ontario, Canada). The TBK1/IKKε inhibitor CAY10576 was purchased from Cayman Chemical (Ann Arbor, MI).

### Cell culture and transfection

3T3-L1 fibroblasts (American Type Culture Collection, Manassas, VA) were cultured and differentiated as described previously (*Liu et al., 2005*). The cells were routinely used within 7 days after completion of the differentiation process; only cultures in which >90% of cells displayed adipocyte morphology were used. 3T3-L1 adipocytes were transfected on the second day post FBS using Amaxa Cell Line Nucleofector Kit L (Lonza, Houston, TX) according to the manufacturer's protocol. 3T3-L1 adipocytes were serum starved for 12 hr with 0.5% fetal bovine serum (FBS) in Dulbecco's modified eagle medium (DMEM, Invitrogen, Grand Island, NY) prior to TNFα treatments (50 ng/ml unless otherwise noted). 3T3-L1 adipocytes were pre-treated for 1 hr with amlexanox at the given concentrations. Alternatively, 3T3-L1 adipocytes were treated with 50 μM forskolin or 10 μM isoproterenol for 15 min, after a 60 min amlexanox pretreatment. The cells were harvested for total RNA and analyzed by real-time PCR. Cell lysates were resolved on SDS-PAGE and analyzed by immunoblot using the indicated antibodies. HEK293T or Cos-1 cells were cultured to 90% confluence and transfected using Opti-MEM media

(Invitrogen) and 3 µl Lipofectamine 2000 (Invitrogen) per µg DNA according to manufacturer's protocol. Coexpression of IKKε or TBK1 with HA-PDE3B was done using a 2 µg kinase: 1 µg PDE3B ratio of the expression constructs.

## Western analyses

Tissues were homogenized in lysis buffer (50 mM Tris, pH 7.5, 5 mM EDTA, 250 mM sucrose, 1% NP40, 2 mM DTT, 1 mM sodium vanadate, 100 mM NaF, 10 mM $Na_4P_2O_7$, and freshly added protease inhibitor tablet), then incubated for 1 hr at 4°C (*Chiang et al., 2009*). Crude lysates were then centrifuged at 14,000 × *g* for 15 min twice and the protein concentration was determined using BioRad Protein Assay Reagent (Bio-Rad, Hercules, CA). Samples were diluted in sodium dodecyl sulfate (SDS) sample buffer and boiled for 5 min at 95°C. Proteins were resolved by SDS-polyacrylamide gel electrophoresis and transferred to nitrocellulose membranes (Bio-Rad, Hercules, CA). Individual proteins were detected with the specific antibodies and visualized on film using horseradish peroxidase-conjugated secondary antibodies (Bio-Rad, Hercules, CA) and Western Lightning Enhanced Chemiluminescence (Perkin Elmer Life Sciences, Waltham, MA).

## Animals and animal care

Wild-type male C57BL/6 mice were fed a high fat diet consisting of 45% of calories from fat (D12451 Research Diets Inc., New Brunswick, NJ) starting at 8 weeks of age for up to 6 months, while normal diet C57BL/6 controls were maintained on normal chow diet consisting of 4.5% fat (5002 Lab Diet, St. Louis, MO). Animals were housed in a specific pathogen-free facility with a 12-hr light/12-hr dark cycle and given free access to food and water. All animal use was in compliance with the Institute of Laboratory Animal Research Guide for the Care and Use of Laboratory Animals and approved by the University Committee on Use and Care of Animals at the University of Michigan.

## Gene expression analysis

Total RNA was extracted from differentiated 3T3-L1 adipocytes using the RNeasy Kit (Qiagen, Valencia, CA) with a DNase step. The Superscript First-Strand Synthesis System for RT-PCR (Invitrogen, Grand Island, NY) was used with random primers for reverse transcription. Real-time PCR amplification of the cDNA was performed on samples in triplicate with Power SYBR Green PCR Master Mix (Applied Biosystems, Carlsbad, CA) using the Applied Biosystems 7900HT Fast Real-time PCR System. *Adrp* was chosen as the internal control for normalization as its expression was not significantly affected by experimental conditions. Sequences of *Ucp1* primers used in this study are 5'-AGGCTTCCAGTA CCATTAGGT-3' and 5'-CTGAGTGAGGCAAAGCTGATTT-3'. Sequences of *Ikbke* primers used in this study are 5'-ACAAGGCCCGAAACAAGAAAT-3' and 5'-ACTGCGAATAGCTTCACGATG-3'. Data were analyzed using the $2^{-\Delta\Delta CT}$ method (*Livak and Schmittgen, 2001*).

## *In vivo* CL-316,243 treatment

Mice were placed on a high fat diet for 6 months, then after 1 week of daily gavage with vehicle, mice were gavaged with either vehicle or amlexanox (25 mg/kg) daily for 4 days. On the fourth day, mice were injected with CL-316,243 (1 mg/kg) or saline control. For analysis of blood metabolites, serum samples were collected before and 15 min after the injection, via a submandibular bleed. Mice were euthanized and WAT samples were collected for cAMP measurement and western blot analysis, 15 min or 20 min after injection. Analysis of oxygen consumption was performed in metabolic cages, as previously described (*Reilly et al., 2013*), by the University of Michigan Metabolic Phenotyping Core. Relative oxygen consumption was obtained by normalization of oxygen consumption rates, after the CL-316,243 injection, to the oxygen consumption rates on day 3 in the same mouse after saline injection. Both injections were performed at 11 am.

## *Ex vivo* glycerol release lipolysis assay in white adipose tissue

For experiments involving CL-316,243, pieces were pre-incubated for 30 min with amlexanox (100 µM) or DMSO vehicle control; then the tissue pieces were transferred to fresh media with and without 10 mM CL-316,243, and media samples were collected every 15 min for 1 hr. To measure glycerol release, 10 µl of supernatant was combined with 200 µl of Free Glycerol Reagent from the Free Glycerol Determination Kit (Sigma) and allowed to incubate for 15 min at room temperature. Absorbance at 540 nm was measured to determine glycerol content and was normalized to determine glycerol production per mg of white adipose tissue.

## Glycerol release lipolysis assay in 3T3-L1 adipocytes

3T3-L1 adipocytes were incubated in DMEM (Invitrogen) without phenol red for 2 hr at 37°C. The cells then were incubated for 90 min at 37°C in HBSS-2% fatty acid-free BSA with 10 µM isoproterenol or 10 µM CL-316,243. Free glycerol concentration was measured by reacting 25 µl of conditioned media with 200 µl Free Glycerol Reagent (Sigma) and absorbance was measured at 540 nm using the manufacturer's protocol. Glycerol release was normalized to cellular protein content.

## cAMP enzyme immunoassay

3T3-L1 adipocytes were treated with β-agonists and/or phosphodiesterase inhibitors and allowed to incubate for the indicated amount of time at 37°C. The cells were lysed with 150 µl 0.1 M HCl, scraped and spun down. A cAMP Enzyme Immunoassay Kit (Sigma CA201) was used to quantify cAMP levels. 50 µl of cell lysates was combined with 50 µl Assay Buffer 2 in each well and cAMP levels were assayed according to the manufacturer's protocol. Tissue samples were homogenized in 5% TCA, then extracted with water-saturated ether, and dried before resuspension in Assay Buffer 2.

## Glosensor cAMP assay in 3T3-L1 adipocytes

80 µl of packed 3T3-L1 cells was electroporated with 3 µg of Glosensor 22-F using Amaxa Cell Line Nucleofector Kit L (Lonza) according to the manufacturer's protocol. Electroporated cells were resuspended in 10 ml of L1-FBS media and 200 µl per well was plated in six columns of an opaque, white, 96-well tissue culture plate (BD Bioscience, San Jose, CA). After 20 hr, media were changed to 100 µl DMEM with 1.5 mg/ml luciferin and allowed to equilibrate for 1 hr. The cells were treated with 50 µM forskolin and luminescence was read every 30 s for 75 min.

## *In vitro* kinase assays

*In vitro* kinase assays were performed by incubating purified kinase (IKKε, TBK1, IKKβ, PKA, or AKT) in kinase buffer containing 25 mM Tris (pH 7.5), 10 mM MgCl2, 1 mM DTT, and 10 µM ATP for 30 min at 30°C in the presence of 0.5 µCi γ-[$^{32}$P]-ATP and 1 µg myelin basic protein (MBP) per sample as a substrate. IKKε and TBK1 were fused to MBP (maltose binding protein) and these fusion proteins were purified from insect SF9 cells by baculovirus expression system by Dr Stuart J Decker (Life Sciences Institute, University of Michigan). IKKβ, PKA, and AKT kinases were purchased from Millipore (Billerica, MA). The kinase reaction was stopped by adding 4X sodium dodecyl sulfate (SDS) sample buffer and boiling for 5 min at 95°C. Supernatants were resolved by SDS-polyacrylamide gel electrophoresis, transferred to nitrocellulose, and analyzed by autoradiography using a Typhoon 9410 phosphorimager (GE Life Sciences, Piscataway, NJ). The bands were quantified using ImageQuant.

## Calf-intestinal phosphatase dephosphorylation

Calf intestinal phosphatase (CIP) was obtained from New England Biolabs (Ipswich, MA). Immunoprecipitated PDE3B was incubated for 1 hr at 37°C in a 100 µl reaction containing 50 mM Tris pH 7.5, 150 mM NaCl, 1% NP-40, EDTA-free protease, and 5 µl CIP.

## Protein purification

For assays requiring soluble protein, purified GST-14-3-3β protein was eluted from glutathione beads by washing beads with 10 mM glutathione in PBS, pH 8.0. The elution was monitored by A280 readings, and fractions containing protein were pooled and dialyzed overnight against 4 L of ice-cold PBS. The proteins were then concentrated using an Amicon centrifugal filtration unit (Millipore). Concentrated proteins were stored at −80°C in PBS containing 10% glycerol and 10 mM DTT.

## GST-14-3-3 Pulldown

For GST-14-3-3 pulldowns, cells were washed twice with ice-cold PBS and then lysed in 1 ml of 14-3-3 pulldown buffer (PD buffer; 15 mM Tris, pH 7.5, 150 mM NaCl, 0.5% NP-40, 1 mM DTT) supplemented with a protease inhibitor tablet (Roche). Lysates were cleared by centrifuging at 13,000×*g* for 10 min and then were incubated with ~10 mg of GST or GST-14-3-3β bound to glutathione beads (GE Healthcare Life Sciences, MI) for 1.5 hr at 4°C. For samples treated with phosphatase, lysates were preincubated with 500 U of calf intestinal phosphatase (New England Biolabs, Inc.) at 37°C for 1 hr before adding GST-14-3-3 beads. The beads were washed three times with 1 ml of PD buffer and then resuspended in 2X SDS sample buffer.

## Preparation of DIG-labeled proteins and overlay assay

GST or GST-14-3-3β was labeled with DIG by incubating 25 mg of protein with 2 ml of 5 mM Digoxigenin-3-O-methylcarbonyl-ε-aminocaproic acid-N-hydroxysuccinimide ester (DIG-NHS; Roche) in 350 ml of PBS for 15 min at room temperature. The labeling reaction was stopped by adding 100 ml of 1 M Tris-HCl, pH 7.4. Labeled protein was dialyzed against 1 L of 25 mM Tris-HCl, pH 7.4 for 1 hr at room temperature, then against 4 L of PBS, pH 7.4 for 4 hrs at 4°C, and finally against 4 L of fresh PBS, pH 7.4 for 16 hrs at 4°C. Labeled protein was then diluted in 25 ml of TBS (50 mM Tris-HCl, pH 7.4, 150 mM NaCl) containing 2 mg/ml BSA (Sigma-Aldrich) and 0.01% sodium azide (Sigma-Aldrich). DIG-labeled proteins were stored at 4°C.

For overlay assays, PDE3B was immunoprecipitated from HEK293T cells and resolved by SDS-PAGE. Proteins were transferred to a nitrocellulose membrane. The membrane was blocked at room temperature overnight in blocking buffer (5% skim milk in TBS-T). The membrane was then incubated with DIG-labeled proteins for at least 4 hr at 4°C and then washed three times with TBS-T. The membrane was then incubated with blocking buffer containing anti-DIG HRP antibody (1:10,000; Roche) for 2 hr at room temperature, and washed three times with TBS-T. Overlays were visualized by reacting with ECL western blotting substrate (Perkin Elmer Life Sciences, Waltham, MA).

## Mass spectrometry

In-gel digestion followed by LC-MS/MS analysis was carried out by the mass spectrometry-based proteomics resource in the Department of Pathology, University of Michigan. Briefly, tryptic peptides were resolved on a nano-C18 reverse phase column and sprayed directly onto Orbitrap mass spectrometer (LTQ-Orbitrap XL, Thermofisher). Orbitrap was operated in a data-dependent mode to acquire one full MS spectrum (resolution of 30,000@400 m/z) followed by MS/MS spectra on six most intense ions (top 6). Proteins were identified by searching data against human protein database (Uniprot, rel. 2010-9) using X!Tandem/TPP software suite. Oxidation of Met, carbamidomethylation of Cys and phosphorylation of Ser, Thr, and Tyr were considered as potential modifications (*Maine et al., 2010*).

## Statistical analyses

Averaged values are presented as the mean ± SEM. When comparing two groups, we performed Student's *t* test to determine statistical significance. When more than two groups and two factors were investigated, we first performed a two-way analysis of variance (ANOVA) to establish that not all groups were equal. After a statistically significant ANOVA result, we performed between-group comparisons using the Tukey *post hoc* analysis for comparisons of all means and Sidak for comparisons of within factor main effect means. ANOVA and Tukey/Sidak tests were performed using GraphPad Prism version 6.

## Acknowledgements

We thank J Hung, R Truscott, and B Poirier for excellent technical assistance, and X Peng for maintaining the mouse colony. We thank the members of the Saltiel laboratory for helpful discussions. We thank V Manganiello (NHLBI, NIH) for production and use of PDE3B antibodies and his expert input and M Birnbaum for the human PDE3B cDNA (University of Pennsylvania). We thank K Inoki (University of Michigan) for the human 14-3-3β cDNA. We also acknowledge technical support from the Michigan Diabetes Research and Training Center.

## Additional information

### Funding

| Funder | Grant reference number | Author |
| --- | --- | --- |
| National Institutes of Health | F30DK089687 | Jonathan Mowers |
| National Institutes of Health | RO1DK60591, R24DK090962 | Alan R Saltiel |

The funder had no role in study design, data collection and interpretation, or the decision to submit the work for publication.

## Author contributions

JM, MU, Conception and design, Acquisition of data, Analysis and interpretation of data, Drafting or revising the article, Contributed unpublished essential data or reagents; SMR, Conception and design, Acquisition of data, Analysis and interpretation of data; JS, DL, Acquisition of data, Analysis and interpretation of data; S-HC, LC, Analysis and interpretation of data, Contributed unpublished essential data or reagents; ARS, Conception and design, Analysis and interpretation of data, Drafting or revising the article

## Ethics

Animal experimentation: All animal use was in compliance with the Institute of Laboratory Animal Research Guide for the Care and Use of Laboratory Animals and approved by the University Committee on Use and Care of Animals at the University of Michigan. All of the animals were handled according to approved institutional animal care and use committee (IACUC) protocols (#A3114-01) of the University of Michigan. The protocol was approved by the Committee on the Ethics of Animal Experiments of the University of Michigan (Permit Number: Pro00004673). The University of Michigan is fully accredited by the Association for Assessment and Accreditation of Laboratory Animal Care, International (AAALAC, Intl) and the animal care and use program conforms to the standards of "The Guide for the Care and Use of Laboratory Animals," Revised 1996.

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
