## [Decision Letter]

Thank you for sending your work entitled “Inflammation produces catecholamine resistance in obesity via activation of PDE3B by the protein kinases IKKε and TBK1” for consideration at *eLife*. Your article has been favorably evaluated by a Senior editor, a Reviewing editor, and 2 reviewers.

The Reviewing editor and the two reviewers discussed their comments before we reached this decision, and the Reviewing editor has assembled the following comments to help you prepare a revised submission.

Mowers et al. investigate mechanisms by which inflammation in response to HFD causes catecholamine resistance. These studies build on their prior work demonstrating that obesity-induced inflammation enhances the expression of non-canonical IKKε and TBK1 in adipocytes. Here, they present evidence that these kinases phosphorylate PDE3B, which subsequently leads to reduced cellular cAMP concentrations and a reduction of cAMP dependent pathways including lipolysis. The authors conclude that reduced energy expenditure in obesity may result from reduced sensitivity to sympathetic activation due to inflammation-generated signals in adipose tissue.

Both reviewers were of the opinion that the manuscript reports significant new findings. One reviewer stated, “Overall this is a timely and well-performed study that provides important new mechanistic information on the functional role of the non-canoncial IKKe and TBK1 proteins in diet induced obesity, energy balance and lipid metabolism.” The second reviewer noted, “The paper is well written and comprehensive despite a cornucopia of data.” Both reviewers also raised some substantial concerns that will need to be addressed by additional experiments. One reviewer made the general comment, “While the results on PDE3B phosphorylation are interesting and potentially important, the effects of IKKe and TBK1 on lipolysis are less impressive and require additional consolidation.” Specific comments to be addressed are as follows:

1) Lipolysis was determined as glycerol release per g of adipose tissue. This is problematic because adipose tissue mass and adipocyte size changes in response to high fat feeding. The “decrease” in lipolytic rates (Figure 1A) may result from increased TAG accumulation and adipocyte mass rather than from reduced lipase activities. Glycerol release should be displayed per cell or per DNA content and not tissue mass.

2) Also, glycerol release is a compound value of TAG hydrolysis and re-esterification of glycerol intermediates. In order to determine whether lipase activities are affected, it is necessary to measure their actual activities in in vitro assays.

3) The increase in lipolysis in response to ISO and FSK are surprisingly moderate (Figure 1A) and glycerol release data correlate relatively poorly to P-HSL levels (comparing Figure 1A with Figure 1C).

4) To strengthen the data on lipolysis the authors should analyze perilipin phosphorylation and the expression of ATGL/CGI-58. Particularly perilipin phosphorylation by PKA is a strong determinant of lipolysis.

5) Figure 2A: Why does UCP1 expression not return to the level of empty vector when IKKe is mutated (IKKe K38A)? What happens to UCP1 expression in TNFa treated 3T3-L1 cells?

6) Throughout all figures the induction of HSL phosphorylation in response to ISO and FSK is very inconsistent. Additionally, glycerol release and cellular cAMP concentrations do not correlate well with HSL phosphorylation (e.g., Figure 4B and Figure 4D) in the TNF experiments – why? What are the long-term TNF-treated phosphorylation patterns of HSL and perilipin?

7) Figure 4C lacks a true negative control of a kinase that does not phosphorylate PDE3B.

8) In several instances, the sample number is insufficient to provide statistically meaningful information. For example, the authors mentioned that in Figure 1A isoproterenol-induced glycerol release in HFD is not significant due to small sample size (n=2). The sample numbers need to be increased in order to make defined conclusions.

9) Isoproterenol induced glycerol release in ND IKKε KO is significantly enhanced compared to ND IKKε WT (Figure 1A). However there were no differences in phospho-PKA substrate phosphorylation (Figure 1B) and relative pHSL/HSL ratio (Figure 1C). In contrast isoproterenol-induced phospho-PKA substrate phosphorylation (Figure 1B) and relative pHSL/HSL ratio (Figure 1C) were significantly increased in HFD IKKε KO compared to HFD IKKε WT, although there was no difference in isoproterenol-induced glycerol release in HFD IKKε KO and HFD IKKε WT (Figure 1A). How can the authors interpret these apparent inconsistencies? Is this due to the lack of statistical significance due to the low sample size?

10) To study β-adrenergic signaling several agonist were selectively employed isoproterenol, forskolin, CL-316,243. It would have been preferable if more consistency in the agonist used were reported. For example, either isoproterenol-induced UCP1 expression or CL-316,243-induced glycerol release should be provided in either Figure 2A or Figure 2B, respectively, for completeness.

11) The data presented in Figure 3 clearly demonstrates that TNF-α induced IKKε in adipocytes suppresses cAMP and glycerol production, and that the IKKε and TBK1 inhibitors amlexanox and CAY10576 rescue the TNF-α induced defects. Since these are also pharmacological manipulations, it would be nice to have a genetic conformation, such as examining the effect of IKKε and TBK1 knockdown on TNF-α-mediated cAMP and glycerol production.

---

## [Author Response]

*1) Lipolysis was determined as glycerol release per g of adipose tissue. This is problematic because adipose tissue mass and adipocyte size changes in response to high fat feeding. The “decrease” in lipolytic rates (Figure 1A) may result from increased TAG accumulation and adipocyte mass rather than from reduced lipase activities. Glycerol release should be displayed per cell or per DNA content and not tissue mass*.

The reviewers bring up a good point, which reveals a fundamental problem in comparing these mouse models in which the weights are different. Although ?-adrenergic signaling and lipolysis are enhanced in white adipose tissue from IKK? knockout mice fed a high-fat diet, the body weights of IKK? knockout mice are reduced compared to WT mice. We realized that conclusions derived from these data are thus confusing and somewhat problematic. Because weight loss alone has the potential to improve catecholamine sensitivity in obese mice, it is impossible to determine from these data whether the differences were due to weight or direct IKK? effects. Furthermore, we now know that in addition to IKKε TBK1 also plays an important role in the regulation of the lipolytic pathway.

Because of these problems in interpretation of the data, we have elected to remove Figure 1 in its entirety from the revised paper. However, in order to address the issue of the direct role of IKK? and TBK1 in energy expenditure and catecholamine sensitivity in vivo, we have added another experiment (Figure 5 of the revised paper), demonstrating that weight-matched, amlexanox-treated mice are more sensitive to CL injection regarding energy expenditure, as measured by indirect calorimetry. We hope this is a more accurate and interpretable experiment that sheds light on this issue with in vivo data.

*2) Also, glycerol release is a compound value of TAG hydrolysis and re-esterification of glycerol intermediates. In order to determine whether lipase activities are affected, it is necessary to measure their actual activities in in vitro assays*.

While it is relatively easy to demonstrate changes in lipolysis after treatment of adipocytes with beta-adrenergic agonists, the corresponding increase in lipase activity as measured in vitro is rarely more than 1.5 fold. Even when HSL is phosphorylated in vitro with recombinant PKA, the change in catalytic activity is relatively modest (up to about 50% increase in most studies (Krintel et al., *FEBS J*, 2009) and many others). The reason for this discrepancy is that in addition to its modest effect on catalytic activity, HSL (and perilipin) phosphorylation induces a translocation of the enzyme to the lipid droplet, where it can gain access to its substrate. This translocation effect relies on phosphorylation, and is probably the more important event in controlling activity in cells (Clifford et al., *JBC*, 2000, and many other papers). Because of this, HSL activity per se is rarely measured in vitro after cell lysis.

We appreciate that in addition to HSL phosphorylation, other events including re-esterification of glycerol intermediates may occur that contribute to changes in overall net lipolysis, and have modified the paper to include these possibilities. However, since the study is largely focused on the effects of IKK?/TBK1 on cAMP signaling, we feel that our demonstrated changes in HSL, p38 and perilipin (see below) phosphorylation are sufficient to make the point.

*3) The increase in lipolysis in response to ISO and FSK are surprisingly moderate (Figure 1A) and glycerol release data correlate relatively poorly to P-HSL levels (comparing Figure 1A with Figure 1C)*.

Because of the reasons explained above, we have removed Figure 1.

*4) To strengthen the data on lipolysis the authors should analyze perilipin phosphorylation and the expression of ATGL/CGI-58. Particularly perilipin phosphorylation by PKA is a strong determinant of lipolysis*.

As requested, we examined levels of phospho-perilipin using an anti-phospho-perilipin antibody that recognizes Ser522, which is a site phosphorylated by PKA (Figures 1, 2 and 5). Perilipin phosphorylation correlated well with HSL phosphorylation. We also analyzed the expression levels of ATGL and CGI-58 in response to long-term TNFα treatment in Figure 2. We observed decreased levels of HSL and perilipin, which are targets of PPARγ (Deng et al., *Endocrinology*, 2006, Arimura et al., *J Biol Chem.*, 2004) because TNF? treatment resulted in a dose-dependent decrease in PPARγ expression in 3T3-L1 adipocytes, as previously reported (Zhang et al., *Mol Endocrinol.*, 1996). However, expression of ATGL and CGI-58 were not changed in response to long-term TNFα treatment.

*5) Figure 2A: Why does UCP1 expression not return to the level of empty vector when IKKe is mutated (IKKe K38A)? What happens to UCP1 expression in TNFa treated 3T3-L1 cells*?

This is an interesting question and points to the limits in interpreting experiments in which the kinase inactive mutants are overexpressed. Firstly, the levels of protein achieved by overexpression of IKK? K38A are consistently less than those seen with the WT protein (Figure 1), probably because the kinase inactive mutant is somewhat toxic to cells. In this regard, using more IKKε K38A DNA in an attempt to reach similar levels of WT protein expression during electroporation was not successful, as the fat cells would not tolerate the transfection. We also tried to generate stable 3T3-L1 adipocytes by lentiviral transduction to resolve this problem, but achieved only 10% infection efficiency in mature 3T3-L1 adipocytes, again reflecting the toxicity produced by the dead kinase. Even though we were not able to achieve equal expression, we repeated our experiments to examine *Ucp1* gene expression in response to β-adrenergic stimuli in 3T3-L1 adipocytes overexpressing empty vector or IKKε, or IKKε K38A by electroporation. Interestingly, as shown before in numerous experiments, overexpression of IKKε K38A still produced a small reduction in the sensitivity of cells to β-adrenergic stimuli regarding *Ucp1* expression (Figure 1), glycerol release (Figure 1) or cAMP signaling (Figure 1) compared to when empty vector was overexpressed, despite the fact that levels of expression were lower than those seen with the WT kinase. We are not able to provide a conclusive explanation for this, but can point to one possibility. TBK1 decreases sensitivity to β-adrenergic stimuli, similar to the effect of IKKε (Figure 1—figure supplement 1), and both endogenous TBK1 and IKKε are present in IKKε K38A expressing cells. Recent studies on crystal structure of TBK1 suggest that homo- and hetero-dimerization of both IKK? and TBK1 (Larabi et al., *Cell rep.*, 2013 and Tu et al., *Cell rep.*, 2013) could result in activation of the complex. Thus, dimerization of IKKε K38A with an endogenous active kinase may produce increased activity of the complex. We have discussed this possibility in the revised paper.

As requested, we also examined the effect of TNF? treatment on *Ucp1* expression in response to isoproterenol or forskolin, or CL-316,243 (Figure 2—figure supplement 1). TNFα treatment reduced the induction of *Ucp1* mRNA levels in response to β-adrenergic stimuli in 3T3-L1 adipocytes.

*6) Throughout all figures the induction of HSL phosphorylation in response to ISO and FSK is very inconsistent. Additionally, glycerol release and cellular cAMP concentrations do not correlate well with HSL phosphorylation (e.g., Figure 4B and Figure 4D) in the TNF experiments – why? What are the long-term TNF-treated phosphorylation patterns of HSL and perilipin*?

We repeated our experiments to examine the effects of long-term TNFα treatment on phosphorylation of HSL and perilipin, and now include this new data in the paper. We observed that phosphorylation of HSL and perilipin correlated well with glycerol release and cellular cAMP levels (Figure 2 compared with Figure 2 and Figure 2 compared with Figure 2 of the revised paper). We also observed that long-term TNFα treatments resulted in a dose-dependent decrease in HSL and perilipin phosphorylation (Figure 2).

*7) Figure 4C lacks a true negative control of a kinase that does not phosphorylate PDE3B*.

As a negative control, we performed in vitro kinase assay using recombinant IKKβ with purified PDE3B as a substrate, since this IKK is most closely related to TBK1 and IKKε. PDE3B phosphorylation was barely detected with IKKβ (Figure 3—figure supplement 1).

*8) In several instances, the sample number is insufficient to provide statistically meaningful information. For example, the authors mentioned that in Figure 1A isoproterenol-induced glycerol release in HFD is not significant due to small sample size (n=2). The sample numbers need to be increased in order to make defined conclusions*.

Because of the reasons explained above, we have removed Figure 1.

*9) Isoproterenol induced glycerol release in ND IKKε KO is significantly enhanced compared to ND IKKε WT (Figure 1A). However there were no differences in phospho-PKA substrate phosphorylation (Figure 1B) and relative pHSL/HSL ratio (Figure 1C). In contrast isoproterenol-induced phospho-PKA substrate phosphorylation (Figure 1B) and relative pHSL/HSL ratio (Figure 1C) were significantly increased in HFD IKKε KO compared to HFD IKKε WT, although there was no difference in isoproterenol-induced glycerol release in HFD IKKε KO and HFD IKKε WT (Figure 1A). How can the authors interpret these apparent inconsistencies? Is this due to the lack of statistical significance due to the low sample size*?

Because of the reasons explained above, we have removed the previous Figure 1.

*10) To study β-adrenergic signaling several agonist were selectively employed isoproterenol, forskolin, CL-316,243. It would have been preferable if more consistency in the agonist used were reported. For example, either isoproterenol-induced UCP1 expression or CL-316,243-induced glycerol release should be provided in either Figure 2 or Figure 2, respectively, for completeness*.

We hoped to show one stimulus at the receptor level (the non-selective β-adrenergic agonist, isoproterenol or the β_3_-adrenergic agonist, CL-316,243) and one beyond the receptor (the adenylyl cyclase activator, forskolin) to emphasize the role of noncanonical kinases IKKε and TBK1 on catecholamine resistance beyond the receptor level. Since differentiated 3T3-L1 adipocytes predominantly expresses the β_3_-adrenergic receptor (Monjo et al., *Am J Physiol Endocrinol Metab.*, 2005), CL-316,243 was also used in some experiments. As requested, we now show isoproterenol or CL-316,243-induced UCP1 expression (Figure 1) and isoproterenol or CL-316,243-induced glycerol release (Figure 1). We view the discovery of post receptor involvement in adrenergic desensitization in obesity and its connection to the noncanonical IκB kinases IKKε and TBK1 as our most significant finding.

*11) The data presented in Figure 3 clearly demonstrates that TNF-α induced IKKε in adipocytes suppresses cAMP and glycerol production, and that the IKKε and TBK1 inhibitors amlexanox and CAY10576 rescue the TNF-α induced defects. Since these are also pharmacological manipulations, it would be nice to have a genetic conformation, such as examining the effect of IKKε and TBK1 knockdown on TNF-α-mediated cAMP and glycerol production*.

This was an excellent idea and we pursued it. Unfortunately, it was not possible to achieve complete knockdown of IKKε and TBK1. Making matters more difficult, TNFα also induces both mRNA and protein levels of IKKε through NFκB activation (Reilly and Chiang et al., *Nat Med.*, 2013), even under knockdown conditions. The data below shows that we get ∼ 80% knockdown of TBK1 and ∼ 50% knockdown of IKKε in 3T3-L1 adipocytes by electroporation, but that 24 hours of TNFα treatment induces IKKε protein levels almost back to control levels, and also decreases PPARγ levels (Zhang et al., *Mol Endocrinol.*, 1996) in a dose dependent manner.Author response image 1.